∂ | **Open Peer Review** | Food Microbiology | Research Article

# Protective effect of *Enterococcus faecium* against ethanol-induced gastric injury via extracellular vesicles

Meiying Luo,[1] Junhang Sun,[1] Suqian Li,[1] Limin Wei,[2] Ruiping Sun,[3] Xin Feng,[1] Huihua Zhang,[1] Ting Chen,[4] Qianyun Xi,[4] Yongliang Zhang,[4] Qien Qi[1]

**ABSTRACT** Recently, *Enterococcus* has been shown to have gastric protective functions, and the mechanisms by which *Enterococcus* modulates gastric function are still being investigated. Herein, we investigated how *Enterococcus faecium* (Efm) and *E. faecium*-derived extracellular vesicles (EVs) (EfmEVs) exert protective effect against ethanol-induced gastric injury by investigating the effect of EfmEVs on gastric mucosal ulcer scoring, histological lesion, mucosal glycoprotein production, acidity, anti-oxidative function, and inflammatory responses in rat. Pretreatment with Efm showed significant reduction of ethanol-induced gastric injury, as evidenced by the lowering of ulcer index, histological lesion, gastric pH, and inflammatory responses and the enhancement of mucosal glycoprotein production and anti-oxidative function. Further functional studies on three bioactive components [inactivated Efm, EfmEVs (EVs), and EV-free supernatants] of the bacterial culture showed that EVs are mostly responsible for the gastroprotective effect. Moreover, EV secretion is beneficial for the gastroprotective effect of Efm. Hence, EVs mediated the protective effect of Efm against ethanol-induced gastric injury by lowering inflammatory responses and enhancing anti-oxidative function and may be a potent anti-inflammatory and anti-oxidative strategy to alleviate hyperinflammatory gastrointestinal tract conditions.

**IMPORTANCE** This study indicated that *Enterococcus faecium* provided a protective effect against rat gastric injury, which involved improvement of the mucosal glycoprotein production, anti-oxidative function, and inflammatory responses. Furthermore, we confirmed that three bioactive components (inactivated Efm, extracellular vesicles, and EV-free supernatants) of *E. faecium* culture also contributed to the gastroprotective effect. Importantly, *E. faecium*-derived EVs showed an effective impact for the gastroprotective effect.

**KEYWORDS** *Enterococcus faecium*, extracellular vesicles, gastric injury, anti-inflammatory, anti-oxidative

Gastric ulcer is the most common gastrointestinal disorder that affects 10% of the world population with different etiologies (1). It is mucosal erosions, which is caused by many factors such as nonsteroidal anti-inflammatory drugs (NSAIDs), *Helicobacter pylori* infection, stress, and alcohol, due to the disturbance between the aggressive acid-pepsin secretion and defensive mucosal factors like mucin secretion and cell shedding (2, 3). Long-term drinking or one-time intake of large amounts of alcohol can directly lead to gastric mucosal lesions including gastric ulcer and gastritis (4, 5).

Probiotics are well known for many health effects on the host when consumed, especially on the gastrointestinal tract (6). *Enterococcus faecium* (Efm) is known to be a normal resident in human and animal gastrointestinal tracts (7), with important functions in maintaining gastrointestinal tract health and improving immunity (8–11).

Address correspondence to Qien Qi, qiqien1987@163.com.

Meiying Luo and Junhang Sun contributed equally to this article. Author order were chosen by the contribution to this work.

The authors declare no conflict of interest.

See the funding table on p. 17.

Certain strains of *E. faecium* have been widely used as probiotics in clinical and animal husbandry settings and for fermentation in foods. For example, *E. faecium* is a major ingredient in MedilacVita (Hanmi, Beijing, China), Ecoflor (Walters Health Care, Den Haag, The Netherlands), and Cylactin (Hoffmann-La Roche, Basel, Switzerland), which are used to improve human and animal gastrointestinal health (12). Accumulating evidence from animal models suggests that probiotic microorganisms are promising for preventing gastric injury. *Lactobacillus reuteri* F-9-35 pretreatment attenuates gastric injury in rats by inhibiting oxidative stress and inflammatory response (4). *Lactobacillus plantarum* ZS62 alleviates gastric injury by regulating antioxidant capacity in mice (5). *Lactobacillus rhamnosus* GG pretreatment attenuates acute gastric mucosal injury by increasing the mucosal prostaglandin E2 level and mucin mRNA expression in rats (13). Another important member of the *Enterococcus* genus, *Enterococcus faecalis* EF-2001, exerts a gastroprotective effect by the suppression of MAPKs and NF-κB signaling and consequent reduction of proinflammatory mediators or cytokines (3). The mechanisms by which probiotic microorganisms modulate gastric function are still being investigated.

Extracellular vesicles (EVs) are lipid-bilayer vesicles produced by all domains of life (14), which play important roles in modulation of immune, inflammatory, regenerative, and remodeling processes by the transfer of the unique compositions to neighboring or distant recipients (15–18). In eukaryotes, EVs are classified into three subtypes: microvesicles, exosomes, and apoptotic bodies based on their size and origin in the cell (19). Similarly, both Gram-positive and Gram-negative bacteria produce EVs, which are referred to as membrane vesicles and outer membrane vesicles, respectively, based on their proposed mechanisms of release (20). For clarity, we refer to all extracellular vesicles as EVs in this paper. Recently, increasing studies have revealed the function of probiotic microorganism-derived EVs in improving gastrointestinal health in animals. *Lactobacillus paracasei*-derived EVs attenuate the intestinal inflammatory response by augmenting the endoplasmic reticulum stress pathway (21). *Lactobacillus reuteri*-derived EVs participated in maintaining intestinal immune homeostasis against LPS-induced inflammatory responses in a chicken model (22). However, the role of EVs, especially probiotic microorganism-derived EVs in gastroprotective effect has not been reported.

In this study, we investigated how Efm and *E. faecium*-derived EVs (EfmEVs) exert gastroprotective effect on gastric mucosal ulcer scoring, histological lesion, mucosal glycoprotein production, acidity, anti-oxidative function, and inflammatory responses in rat.

## MATERIALS AND METHODS

### Identification and culture of bacteria

Efm was isolated from a commercial microecologics used in animal husbandry. Species identification was performed using 16S ribosomal RNA (rRNA) gene sequencing by Shanghai Sangon Biotech as previously described (Table S1) (23). Liquid cultures were grown in MRS medium (24) (2.0% glucose, 1.0% meat extract, 1.0% peptone, 0.5% yeast extract, 0.5% sodium acetate, 0.2% $K_2HPO_4$, 0.2% diammonium citrate, 0.1% Tween 80, 0.02% $MgSO_4 \cdot 7H_2O$, and 0.005% $MnSO_4 \cdot H_2O$). Bacteria were shaken at 200 rpm in aerobic conditions at 37°C for 14～16 h to an optical density (OD)$_{600}$ of 0.4～0.6.

### Inactivation of bacteria

Inactivation of bacteria was performed as previously described (25, 26). Efm cultures were centrifuged at 12,000 rpm for 10 min at 4°C, and the supernatants were discarded. Bacterial deposits were resuspended in phosphate-buffered saline (PBS) after being washed in sterile PBS. Before the bacteria were inactivated, the culture was plated to calculate the live CFU equivalents. Inactivated Efm (iaEfm) were prepared by incubation at 85°C for 30 min after harvest by centrifugation. Agar plating was performed to verify the complete inactivation of bacteria.

## Isolation and characterization of bacterial EVs

EfmEVs were isolated from the cultured supernatants using differential ultracentrifugation (dUC) as previously reported (27). Briefly, culture broth samples were collected and centrifuged at 2,000 × $g$ for 10 min at 4°C and at 10,000 × $g$ for 30 min at 4°C to remove debris. The supernatants from these spins were then centrifuged at 120,000 × $g$ for 90 min at 4°C using SW32 Ti rotor and Optima XPN ultracentrifuge (Beckman Coulter Instruments, Fullerton, CA, USA). The supernatants were harvested as EV-free supernatants (EVFS). The pellets containing EfmEVs were washed by PBS and then centrifuged at 120,000 × $g$ for 90 min at 4°C again. The EfmEV pellets were resuspended in 2 mL of sterile PBS. Agar plating was performed to ensure no bacterial contamination in the purified EfmEVs and EVFS (the EfmEVs and EVFS were sterilized using 0.45-µm filters before use). The EfmEV and EVFS preparations were stored at −80°C until use.

Protein concentrations of the EfmEV and EVFS preparations were determined with a BCA Protein Assay Kit (BCA Protein Assay Kit, P0010; Beyotime Biotechnology, CN). EfmEV particle size distribution was measured by dynamic light scattering (DLS) with ZETASIZER Nano series-Nano-ZS (Malvern Instrument, UK), as previously reported (21). The morphology of EfmEVs was observed by transmission electron microscopy (TEM). Briefly, the purified EfmEVs were applied to a copper grid coated with formvar for 2 min, washed with ultrapure water, negatively stained with 1% uranyl acetate, observed, and photographed by TEM (JEM-2000EX; Jeol, Tokyo, Japan) (21).

## Stimulation of EVs secretion

Linezolid (LZD) is a synthetic antibiotic which prevents the synthesis of bacterial protein via binding to rRNA on both the 30S and 50S ribosomal subunits (28). LZD-induced EV secretion was performed as described previously (29). To confirm whether LZD is capable of stimulating EV secretion by Efm, Efm at the concentration of 1 × $10^5$ CFU/mL were cultured in 40 mL MRS medium containing LZD {0.2 µg/mL [MIC 2 µg/mL (29)]}, HY-10394, MedChemExpress, New Jersey, USA), or an equal volume of DMSO. Bacteria were shaken overnight at 200 rpm in aerobic conditions at 37°C. The conditioned media of Efm were harvested for EfmEV isolation as described above, in order to evaluate the stimulation effects of LZD on EV secretion by Efm. Protein concentrations of the isolated EfmEVs were determined with a BCA Protein Assay Kit. EfmEV particle size distribution was measured by DLS as described above. The total protein contents in EfmEV- and LZD-stimulated EfmEV samples were assessed via sodium dodecyl sulfate-polyacrylamide gel electrophoresis followed by Coomassie Brilliant Blue staining, with a loading amount of 30 µg protein.

## Animals

Sprague Dawley (SD) rats (specified pathogen free (SPF), female, 190 ~ 230 g, 8 weeks of age) and BALB/c mice (SPF, female, 13 ~ 15 g, 4 weeks of age) were obtained from the Guangdong Medical Laboratory Animal Center (Foshan, Guangdong, China). Animals were housed at 22°C ± 2°C in 12 h light/dark cycles and domesticated for a week before the experiment. A standard diet was fed, and water was provided *ad libitum*.

## Experimental design

To explore the protective effect of Efm on ethanol-induced gastric injury, 32 female SD rats were randomly divided into four groups (*n* = 8): normal control group (PBS), ethanol model group (EtOH), positive control group [omeprazole (OME)], and Efm group (Efm). The PBS and EtOH groups were orally administered with 5 mL/kg body weight (BW) of PBS, while the OME and Efm groups received 4 mg/mL of omeprazole (volume 5 mL/kg BW) (3) or 2 × $10^9$ CFU/mL of Efm (volume 5 mL/kg BW) (4, 5) every other day for a total of three times. The animals were fasted for 12 h and allowed to water access before the last treatment. The animals in EtOH, OME, and Efm groups were orally administered with

a single dose of absolute ethanol (5 mL/kg BW) to induce acute gastric mucosal injury at 2 h after the last treatment (4, 30), while the PBS group received 5 mL/kg BW of PBS.

In our experiment, each 1-mL overnight cultures contained about $2 \times 10^9$ CFU Efm and 20 µg EfmEVs. Therefore, 1 mL EVFS was comparable to $2 \times 10^9$ CFU-inactive Efm and 20 µg EfmEVs in the subsequent tests. To determine which fraction of the bacterial cultures is mostly responsible for the gastroprotective effect, we studied the protective effect of different active components of bacterial cultures against gastric injury. Twenty-four female SD rats were randomly divided into three groups ($n = 8$): inactivated Efm group (iaEfm), EfmEV group (EVs), and EVFS group (EVFS). The iaEfm, EV, and EVFS groups were orally administered with $2 \times 10^9$ CFU/mL of iaEfm (volume 5 mL/kg BW), 20 µg/mL of EfmEVs (volume 5 mL/kg BW), or 5 mL/kg BW of EVFS, respectively, every other day for a total of three times. Fasting and ethanol induction were performed as described above.

To further investigate whether EV secretion is beneficial for the gastroprotective effect of Efm, 24 female SD rats were randomly divided into three groups ($n = 8$): LZD group (LZD), Efm group (Efm), and Efm + LZD-cotreated group (Efm-LZD). The LZD, Efm, and Efm-LZD groups were orally administered with 0.2 µg/mL of LZD (volume 5 mL/kg BW), $2 \times 10^9$ CFU/mL of Efm (volume 5 mL/kg BW), and Efm mixed with LZD ($2 \times 10^9$ CFU/mL of Efm, 0.2 µg/mL of LZD, and volume 5 mL/kg BW) every other day for a total of three times. Fasting and ethanol induction were performed as described above.

## Necropsy and gastric mucosal ulcer scoring

All animals were sacrificed under anesthesia after injection of 1% pentobarbital sodium (50 mg/kg BW) 2 h after ethanol induction. The stomach was removed, opened along the greater curvature, and rinsed with phosphate-buffered saline ($1 \times$ PBS, 4°C). The stomach was stretched on a clean paper with the mucosal surface facing upward. Photographs of gastric mucosal ulcer of the stomach were taken with a Huawei Mate 40 Pro mobile phone camera (Huawei Technologies Co. Ltd., Shenzhen, China). The degree of gastric mucosal damage was graded according to ulcer score scales as described previously (31): 0, no lesions; 1, one hemorrhagic ulcer with length < 5 mm and thin; 2, one hemorrhagic ulcer with length > 5 mm and thin; 3, more than one ulcer grade 2; 4, one ulcer with length > 5 mm and width > 2 mm; 5, two or three ulcers of grade 4; 6, from four to five ulcers of grade 4; 7, more than six ulcers of grade 4; and 8, complete lesion of the mucosa. Mean scores for each group were calculated and expressed as ulcer index (UI).

## Histopathological assessment of gastric damage

The stomach was rinsed with sterile PBS and fixed in 4% paraformaldehyde (vol/vol) for 48 h. Tissues were then dehydrated and embedded in paraffin. Full-thickness sections (5 µm) were prepared and then stained by hematoxylin and eosin (H&E) to evaluate gastric histological damage and stained with periodic acid Schiff's (PAS) to evaluate mucosal glycoprotein production as described previously (2). Sections were photographed using a Nikon Eclipse E200 microscope (Nikon, Tokyo, Japan).

## Gastric acidity (pH) and biochemical analysis

The pH of the gastric contents was determined by a digital pH meter. The gastric tissues were homogenized (1/10; wt/vol) in $1 \times$ PBS (4°C), and the supernatants were collected after centrifugation at 12,000 rpm for 15 min. The levels of H+/K+-ATPase, glutathione peroxidase (GSH-Px), glutathione (GSH), malondialdehyde (MDA), and myeloperoxidase (MPO) in gastric tissue were determined using commercial kits (Jiancheng Bioengineering Institute, Nanjing, China) according to the manufacturer's instructions. Total protein content was measured with a Bradford Protein Assay Kit (Beyotime, Shanghai, China).

Serum was obtained by centrifuging the blood at 12,000 rpm for 15 min. The serum levels of GSH-Px, GSH, MDA, nitric oxide (NO), MPO, tumor necrosis factor-α (TNF-α), interleukin-1β (IL-1β), and interleukin-10 (IL-10) were tested with corresponding kits

(Jiancheng Bioengineering Institute, Nanjing, China). Endothelin-1 (ET-1) was tested with the corresponding kit (Shanghai Enzyme-linked Biotechnology Co. Ltd., Shanghai, China).

## Measurements of mRNA expression in the gastric tissues

Rat gastric samples were preserved with RNAlater (Ambion) and stored at −80℃. Total RNA was isolated using TRIzol reagent (Invitrogen) following the manufacturer's instructions. Reverse transcription was performed with 1 µg of total RNA using the RT EasyTM II cDNA Synthesis Kit (Foregene, Chengdu, China) to synthesize the first-strand cDNA. Target-specific primers were designed using the program Primer5 and are shown in Table S2. Real-time PCR was carried out using SYBR Green Supermix (Foregene, Chengdu, China) on a quantStudio3 (Applied Biosystems) machine. Amplification data were analyzed using the $2^{-\Delta\Delta Ct}$ method with the *β-actin* gene serving as reference.

## Biodistribution of EfmEVs

To determine the biodistribution of EfmEVs, the EVs were labeled with the NIR membrane dye Dir (1,1-dioctadecyl-3,3,3,3-tetrametylindotricarbocyanine iodide) (#UR21017, Umibio, Shanghai, China) as described previously (32). Balb/c mice were orally administered Dir-labeled EVs (1,000 µg/100 µL), equal number of EVs, or equal number of free Dir. After 0 h, 3 h, 6 h, or 12 h, the fluorescence was monitored using the IVIS Spectrum Xenogen machine (Caliper Life Sciences, Hopkinton, MA), after which the mice were immediately sacrificed and the organs were harvested to analyze fluorescence distribution. The signal of Dir-labeled EVs in murine tissues was compared with those produced by an equal number of EVs and free Dir. Only when the signal of Dir-labeled EVs was significantly greater than signals produced by free Dir, tissues were considered potential sites of EV accumulation (33).

## Statistical analysis

The experimental data were analyzed with SPSS 17.0 (SPSS Inc., Chicago, IL, USA). The results are expressed as the mean ± standard deviation. Statistical significance was determined by one-way Analysis of Variance (ANOVA) for three or more groups or Student's *t*-test for two groups. $P < 0.05$ was considered significant.

## RESULTS

### Orally administration of Efm alleviated ethanol-induced gastric injury

### *Efm pretreatment reduced gastric mucosal ulcer, histological lesions, and mucosal glycoprotein lesion*

The schematic diagram of the experimental procedures was shown in Fig. 1a. SD rats were pretreated with Efm every other day for a total of three times to determine the prevention impact of Efm on ethanol-induced gastric injury. Macroscopic evaluation of gastric injury was shown in Fig. 1b. The results showed stomachs from the ethanol model group had severe bleeding and ulcer. Pretreatment with OME or Efm relieved gastric damage compared with the EtOH group, and the Efm group showed the smaller injury. Compared with the EtOH group, UI in the Efm group was significantly decreased (UI, 5.57 ± 1.30 vs. 6.88 ± 0.79, $P < 0.05$), while there were no significant differences in either the OME or Efm group ($P > 0.05$). Histopathological analysis of gastric mucosa stained with H&E was shown in Fig. 1c. PBS group micrograph depicted normal gastric mucosal histology. EtOH group micrograph indicated severe pathological changes including bleeding (✗) and epithelial damage (★). OME group micrograph indicated moderate pathological changes. Efm group micrograph indicated mild pathological changes. Sections were also stained with PAS (Fig. 1d) to evaluate mucosal glycoprotein production. PBS and Efm groups showing strong reaction among surface mucous and neck cells (▲). EtOH and OME groups showing faint PAS reaction. Considered together, these

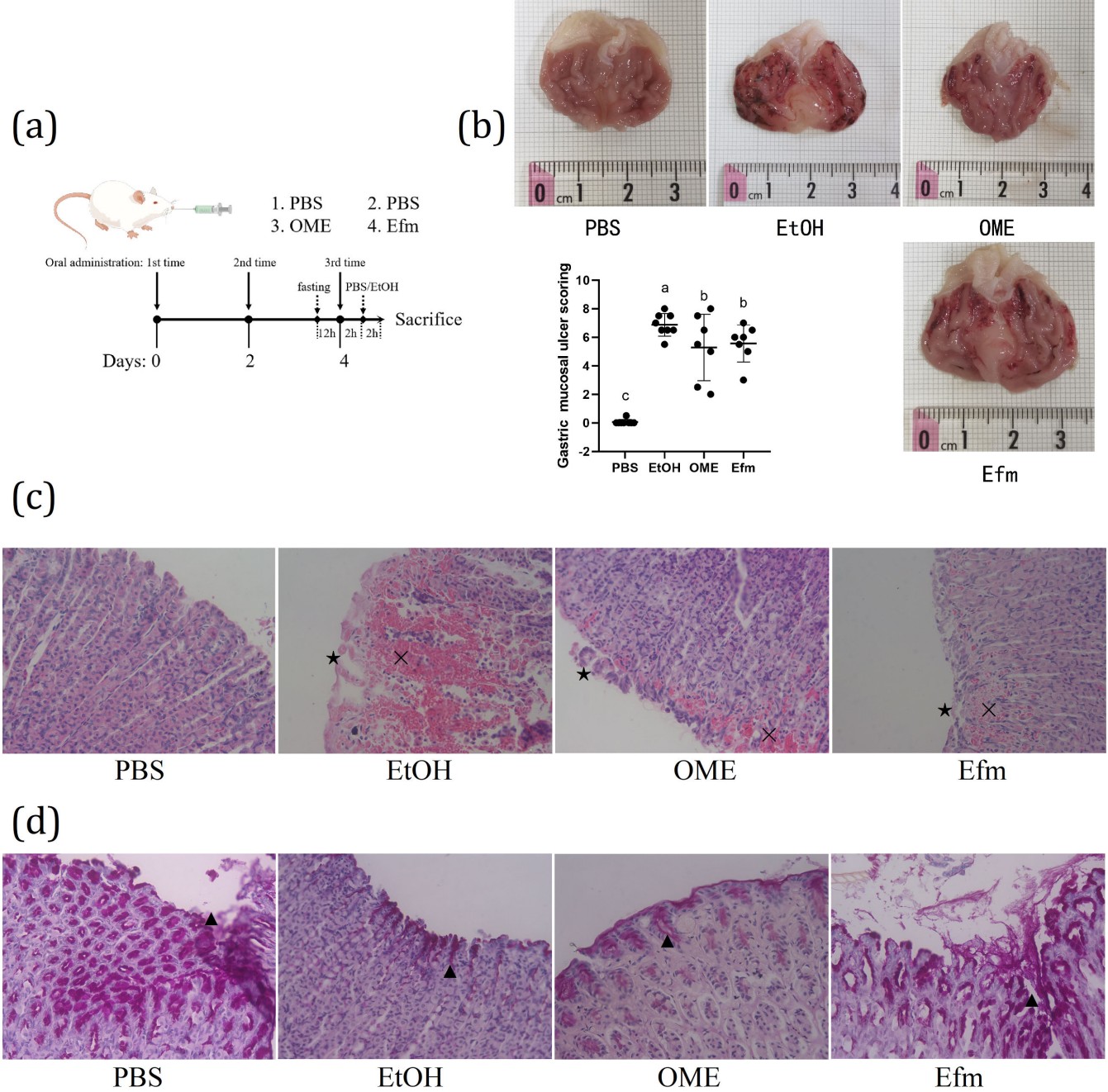

**FIG 1** Pretreatment with Efm attenuated ethanol-induced gastric injury in rats. (a) Experimental strategy used for the animal experiments. To explore the protective effect of Efm on ethanol-induced gastric injury, rats were divided into PBS, EtOH, OME, and Efm groups (n = 8/group). The animals in EtOH, OME, and Efm groups were, respectively, gavaged with 5 mL/kg BW of PBS, 4 mg/mL of OME (volume 5 mL/kg BW), or $2 \times 10^9$ CFU/mL of Efm (volume 5 mL/kg BW) every other day for a total of three times. Before the final gavage, rats were fasted for 12 h with free access to water. Two hours after the final gavage, the rats were gavaged with a single dose of absolute ethanol (5 mL/kg BW) to induce acute gastric mucosal injury. The rats in the PBS group were gavaged with 5 mL/kg BW of PBS every other day for a total of three times. Before the final gavage, rats were fasted for 12 h with free access to water. Two hours after the final gavage, the rats were gavaged with 5 mL/kg BW of PBS instead of ethanol. (b) Macroscopic evaluation of gastric injury. The bar graphs show gastric mucosal ulcer index determined by morphological analysis. (c) Histopathological analysis of gastric mucosa (H&E stain, ×400). (d) Photomicrographs showing different reactions among mucosal layer in stomach sections in different groups (PAS stain, ×400). Data are expressed as mean ± SD. Means with different letters were significantly different (n = 8; one-way ANOVA; Duncan; $P < 0.05$).

results suggested that pretreatment with Efm could significantly ameliorate ethanol-induced gastric injury phenotypes.

## Efm pretreatment recovered gastric pH, H+/K+-ATPase activity, and antioxidant levels

To further illuminate how Efm pretreatment remitted gastric injury severity, gastric pH, H+/K+-ATPase activity, and antioxidant levels were assessed. As shown in Fig. 2a, ethanol treatment significantly increased H+/K+-ATPase, MDA, and MPO levels in gastric tissues ($P < 0.05$) compared with the PBS group. Pretreatment with OME relieved H+/K+-ATPase, MDA, MPO levels compared with the EtOH group ($P < 0.05$), while gastric pH was significantly increased ($P < 0.05$) and the GSH level in gastric tissues was clearly reduced ($P < 0.05$) compared with the PBS group. After Efm pretreatment, gastric pH and H+/K+-ATPase activity levels in gastric tissues were markedly improved compared with the OME group ($P < 0.05$) and MDA and MPO levels were markedly improved compared with the EtOH group ($P < 0.05$). Gastric pH, H+/K+-ATPase, GSH-Px, GSH, MDA, and MPO levels in gastric tissues all recovered to a similar level as in the PBS group ($P > 0.05$). These results suggested that Efm pretreatment ameliorated ethanol-induced gastric injury via the inhibition of gastric acid secretion and the enhancement of the mucosal antioxidant level.

## Efm pretreatment improved gastric mucin secretion proteins and inflammatory-related marker mRNA expression

We next quantified the expression of vasoconstrictor (ET-1), mucin secretion proteins (Muc1, Muc6), and inflammatory related markers (Nfkb1, IL-1b, IL-6, and IL-10) in the gastric samples by reverse transcription-polymerase chain reaction (RT-PCR). As shown in Fig. 2b, ethanol treatment significantly reduced mucin 1 (Muc1) and mucin 6 (Muc6) ($P < 0.05$) and increased interleukin 1 beta (IL-1b), interleukin 6 (IL-6), and interleukin 10 (IL-10) mRNA expression ($P < 0.05$) compared with the PBS group. OME pretreatment relieved IL-1b and IL-6 mRNA expression compared with the EtOH group ($P < 0.05$), while

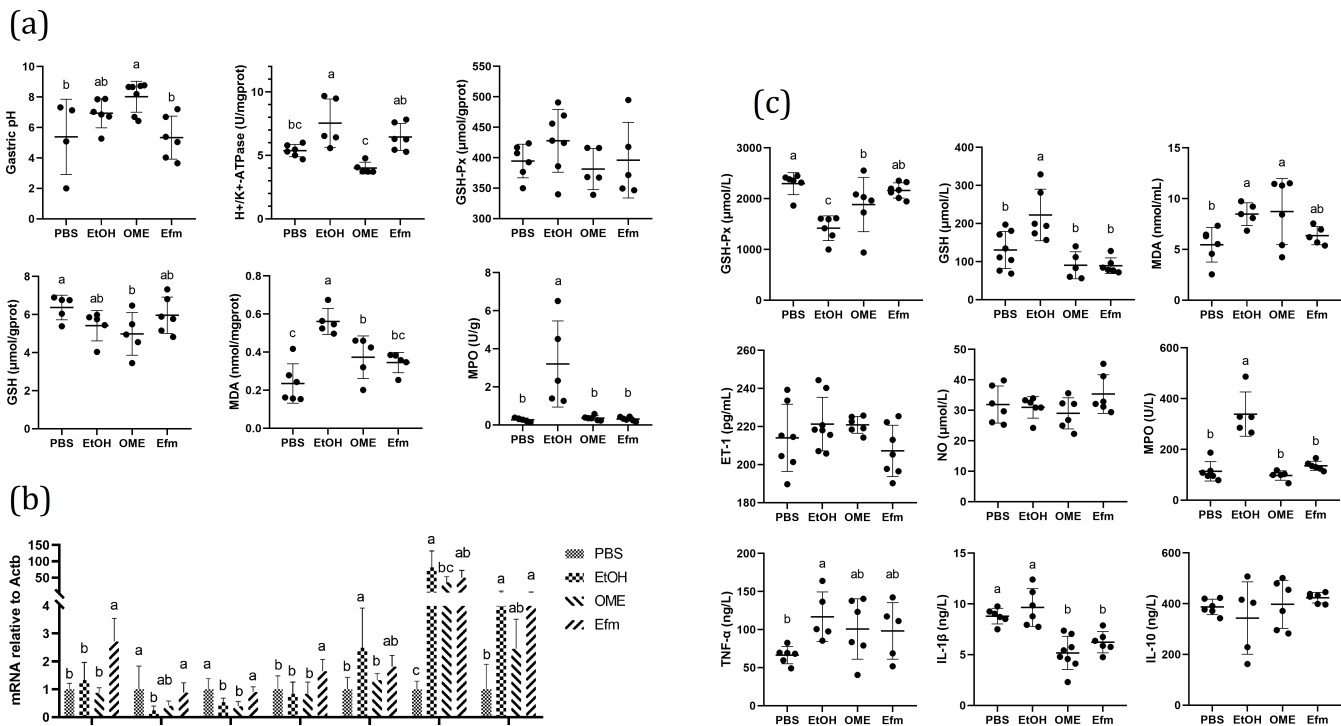

**FIG 2** Effects of pretreatment with Efm on gastric pH, biochemical indexes, and mRNA expression in rat. (a) Gastric pH and the contents of H+/K+-ATPase, GSH-Px, GSH, MDA, and MPO in gastric tissues. (b) Relative mRNA expression levels of *ET-1*, *Muc1*, *Muc6*, *Nfkb1*, *IL-1b*, *IL-6*, and *IL-10* in the gastric tissues. (c) The contents of GSH-Px, GSH, MDA, ET-1, NO, MPO, TNF-α, IL-1β, and IL-10 in serum. Data are expressed as mean ± SD. Means with different letters were significantly different ($n = 8$; one-way ANOVA; Duncan; $P < 0.05$).

the Muc6 mRNA level was significantly reduced ($P < 0.05$) compared with the PBS group. After Efm pretreatment, the Muc6 mRNA level was markedly improved compared with the OME group ($P < 0.05$); ET-1 and nuclear factor kappa B subunit 1 (Nfkb1) mRNA levels were significantly increased compared with the OME group ($P < 0.05$); ET-1, Muc1, Muc6, and Nfkb1 mRNA levels were significantly increased compared with the EtOH group ($P < 0.05$); IL-1b, IL-6, and IL-10 mRNA expression was maintained at a similar level as in the EtOH group ($P > 0.05$); ET-1, Nfkb1, IL-6, and IL-10 mRNA levels were significantly increased compared with the PBS group ($P < 0.05$); Muc1, Muc6, and IL-1b mRNA levels all recovered to similar levels as in the PBS group ($P > 0.05$). These results suggested that Efm pretreatment ameliorated ethanol-induced gastric injury via the enhancement of mucosal barrier function and the inhibition of the proinflammatory response.

### Efm pretreatment recovered the serum antioxidant level and inflammatory factor characteristics

We further assessed the serum antioxidant level and inflammatory factor characteristics. As shown in Fig. 2c, ethanol treatment significantly reduced GSH-Px content ($P < 0.05$) and increased GSH, MDA, MPO, and TNF-α contents in serum ($P < 0.05$) compared with the PBS group. OME pretreatment relieved GSH-Px, GSH, MPO, and TNF-α contents compared with the EtOH group ($P < 0.05$), while MDA content was significantly increased ($P < 0.05$) and GSH-Px and IL-1β contents were clearly reduced ($P < 0.05$) compared with the PBS group. After Efm pretreatment, there were no significant difference in GSH-Px, GSH, MDA, ET-1, NO, MPO, TNF-α, IL-1β, and IL-10 contents compared with the OME group ($P > 0.05$); GSH-Px, GSH, MDA, MPO, TNF-α, and IL-1β contents were markedly improved compared with the EtOH group ($P < 0.05$). The contents of GSH-Px, GSH, MDA, ET-1, NO, MPO, TNF-α, and IL-10 all recovered to a similar level as in the PBS group ($P > 0.05$), and IL-1β content was clearly reduced compared with the PBS group ($P < 0.05$). Results suggested that Efm pretreatment ameliorated ethanol-induced gastric injury via the enhancement of the serum antioxidant level and the inhibition of the proinflammatory response.

### EVs is an important functional active component of Efm

### Characterization and biodistribution of EfmEVs

EfmEVs were isolated from culture supernatants using conventional dUC. DLS and TEM were used to characterize the size and morphology of the EfmEVs. In the DLS measurements, there were two distinct populations of particle size; the main peak was 188.5 nm; EVs showed a slight variation in diameter ranging from 2 to 1,281 nm, and the average size was 94.34 nm (Fig. 3a). TEM images showed the spherical shape of the EfmEVs, which consisted of a lipid bilayer. An *in vivo* imaging study was performed to evaluate whether EfmEVs moved to target organs, including the stomach, intestine, brain, heart, liver, spleen, lung, and kidneys, after oral administration. Whole-mice imaging showed that EfmEVs were present in the stomach and intestine areas after application and diffused in a time-dependent manner (Fig. 3c). In addition, imaging data of dissected organs showed that EfmEVs moved from the stomach to the intestine immediately after administration and finally disappeared 12 h after application (Fig. 3c).

### EfmEV pretreatment reduced gastric mucosal ulcer, histological lesions, and mucosal glycoprotein

In order to determine which fraction of the bacterial culture is mostly responsible for the gastroprotective effect, we studied the gastroprotective effect of three different active components: iaEfm, EfmEVs, and EVFS. The schematic diagram of the experimental procedures was shown in Fig. 4a. SD rats were pretreated with iaEfm, EfmEVs, or EVFS every other day for a total of three times before ethanol treatment. Macroscopic evaluation of gastric injury was shown in Fig. 4b. The results showed stomachs from the EV group showed the smallest injury. Compared with the iaEfm group, UI in the EV group

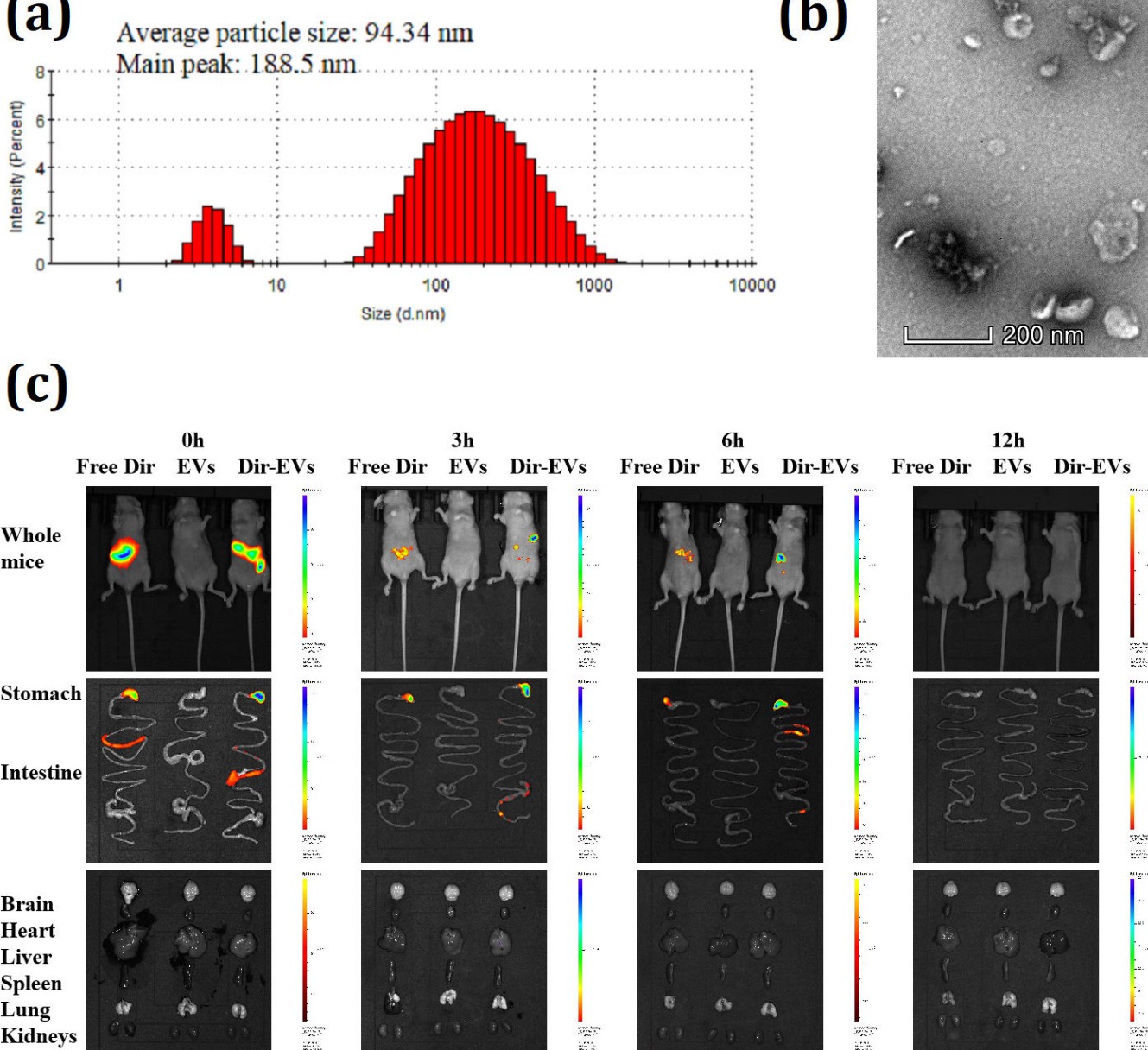

**FIG 3** Characterization and biodistribution of EfmEVs. (a) The mean particle diameter and main peak of EfmEVs, as measured using DLS. (b) Morphology of EfmEVs under a TEM (scale bar = 200 nm). (c) The biodistribution of Dir-labeled EfmEVs in the mouse stomach and intestine was detected *in vivo* and *ex vivo* via fluorescent imaging after oral administration of free Dir, EVs, and Dir-EVs for 0 h, 3 h, 6 h, and 12 h.

was significantly decreased (UI, 4.14 ± 2.23 vs. 6.25 ± 1.00, $P < 0.05$); there were no significant differences in either the EVFS or the EV group ($P > 0.05$). Histopathological analysis of gastric mucosa stained with H&E was shown in Fig. 4c. IaEfm group micrograph indicated moderate pathological changes including bleeding (✗) and epithelial damage (★). EVFS group micrograph indicated moderate pathological changes including bleeding (✗). EV group micrograph indicated mild pathological changes. Sections were also stained with PAS (Fig. 4d) to evaluate mucosal glycoprotein production. IaEfm and EVFS groups show faint PAS reaction (▲). The EV group shows strong PAS reaction. Considered together, these results suggested that EVs are the most responsible fraction for the gastroprotective effect.

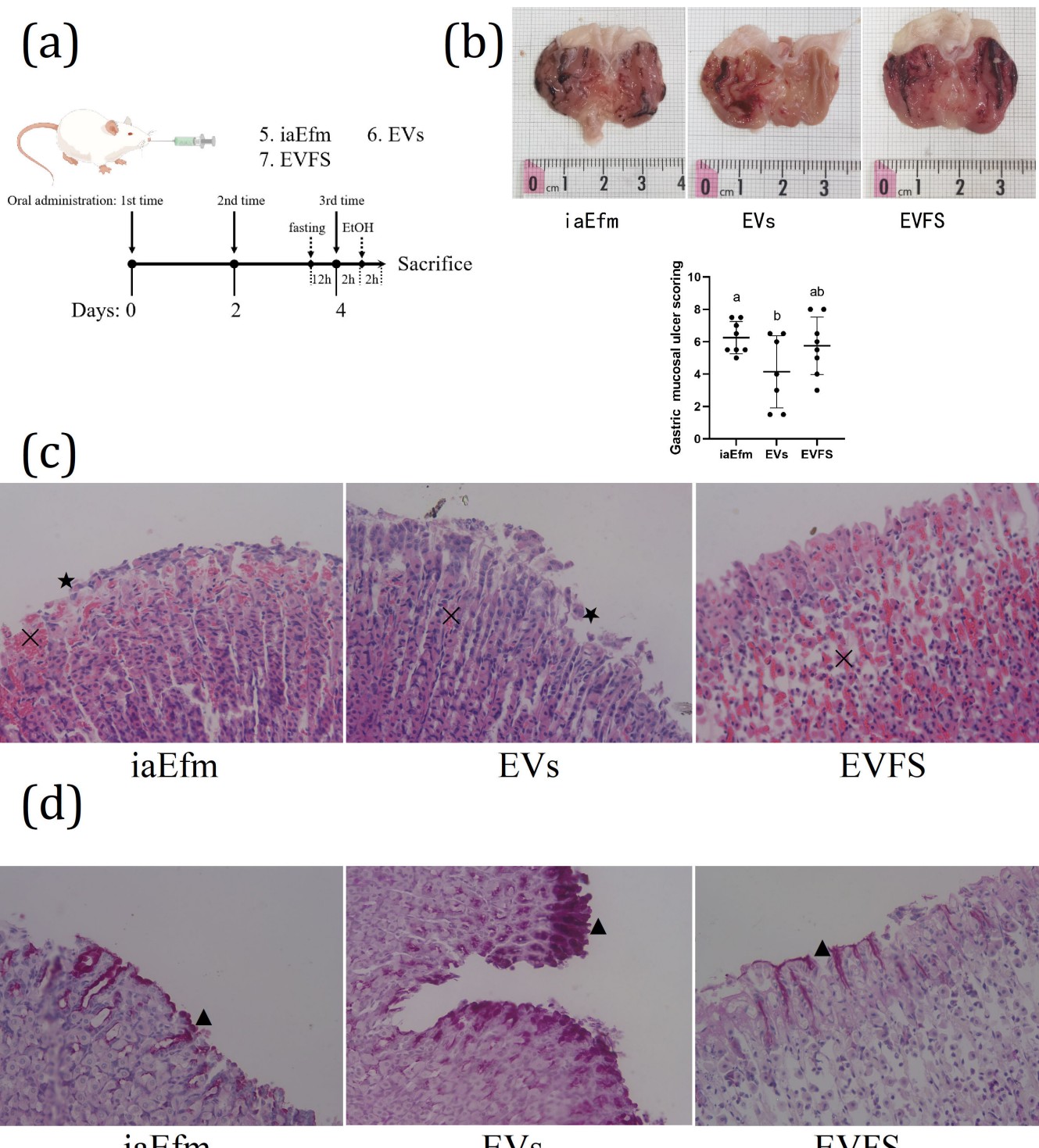

**FIG 4** Protective effect of different active components of bacterial culture against gastric injury in rats. (a) Experimental strategy used for the animal experiments. To determine which fraction of the bacterial culture is responsible of the gastroprotective effect, rats were divided into iaEfm, EV, and EVFS groups (n = 8/group). The rats in the iaEfm, EV, and EVFS groups were gavaged with $2 \times 10^9$ CFU/mL of iaEfm (volume 5 mL/kg BW), 20 μg/mL of EfmEVs (volume 5 mL/kg BW), or 5 mL/kg BW of EVFS, respectively, every other day for a total of three times. Before the final gavage, rats were fasted for 12 h with free access to water. Two hours after the final gavage, the rats were gavaged with a single dose of absolute ethanol (5 mL/kg BW) to induce acute gastric mucosal injury. (b) Macroscopic evaluation of gastric injury. The bar graphs show gastric mucosal ulcer index determined by morphological analysis. (c) Histopathological analysis of gastric mucosa (H&E stain, ×40). (d) Photomicrographs showing different reactions among mucosal layer in stomach sections in different groups (PAS stain, ×400). Data are expressed as mean ± SD. Means with different letters were significantly different (n = 8; one-way ANOVA; Duncan; $P < 0.05$).

### EfmEV pretreatment reduced gastric pH and H+/K+-ATPase activity levels and improved the antioxidant level

We further assessed gastric pH, H+/K+-ATPase activity, and antioxidant levels. As shown in Fig. 5a, EV pretreatment displayed lowest gastric pH, gastric tissue H+/K+-ATPase, GSH-Px, MDA, and MPO levels among three groups. EV pretreatment significantly reduced H+/K+-ATPase and MPO levels in gastric tissues compared with the EVFS group ($P < 0.05$). These results suggested that EV pretreatment ameliorated ethanol-induced gastric injury via the inhibition of gastric acid secretion and the enhancement of the mucosal antioxidant level.

### EfmEV pretreatment improved gastric mucin secretion proteins and inflammatory-related marker mRNA expression

We next quantified the expression of vasoconstrictor (ET-1), mucin secretion proteins (Muc1, Muc6), and inflammatory related markers (Nfkb1, IL-1b, IL-6, and IL-10) in the gastric samples by RT-PCR. As shown in Fig. 5b. EVs pretreatment displayed highest ET-1, Muc1, Muc6, Nfkb1 and lowest IL-1b, IL-6, IL-10 mRNA expression among three groups. No significant differences were found between groups ($P > 0.05$). These results suggested that EVs pretreatment ameliorated ethanol-induced gastric injury via the enhancement of mucosal barrier function and the inhibition of the proinflammatory response.

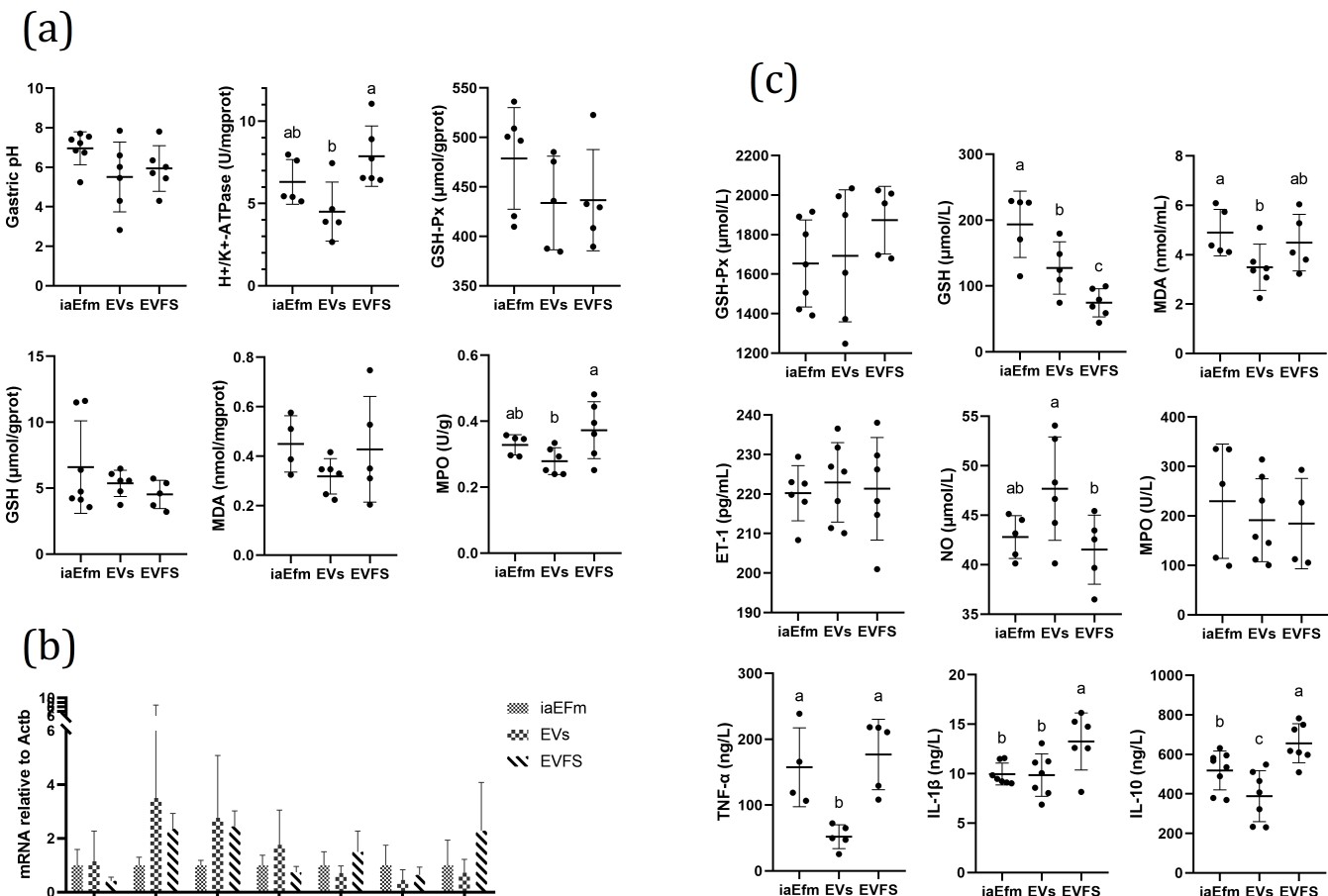

**FIG 5** Effects of pretreatment with Efm different active components on gastric pH, biochemical indexes, and mRNA expression in rat. (a) Gastric pH and the contents of H+/K+-ATPase, GSH-Px, GSH, MDA, and MPO in gastric tissues. (b) Relative mRNA expression levels of *ET-1*, *Muc1*, *Muc6*, *Nfkb1*, *IL-1b*, *IL-6*, and *IL-10* in the gastric tissues. (c) The contents of GSH-Px, GSH, MDA, ET-1, NO, MPO, TNF-α, IL-1β and IL-10 in serum. Data are expressed as mean ± SD. Means with different letters were significantly different ($n = 8$; one-way ANOVA; Duncan; $P < 0.05$).

## EfmEVs pretreatment improved serum antioxidant level and inflammatory factors characteristics

We further assessed serum antioxidant level and inflammatory factors characteristics. As shown in Fig. 5c. EVs pretreatment displayed highest ET-1, NO and lowest MDA, TNF-α, IL-1β, IL-10 contents in serum among three groups. EVs pretreatment significantly reduced serum GSH, MDA, TNF-α, IL-10 contents compared to iaEfm group ($P < 0.05$). EVs pretreatment significantly reduced serum TNF-α, IL-1β, IL-10 contents and increased GSH, NO contents compared to EVFS group ($P < 0.05$). Results suggested that EVs pretreatment ameliorated ethanol-induced gastric injury via the enhancement of serum antioxidant level and the inhibition of the proinflammatory response.

## EVs secretion is beneficial for the gastroprotective effect of Efm

### LZD stimulated EfmEVs secretion

To further investigate whether EVs secretion is beneficial for the gastroprotective effect of Efm, we coincubated Efm with linezolid (LZD), that is able to stimulate the release of EVs at sub-minimum inhibitory concentration (MIC) (29), to interfere with the secretion of EVs by Efm. EVs were isolated from the culture supernatants to determine the effect of LZD on EVs production and their protein profiles. Efm cultured with 1/10 MIC of LZD produced 2.3 (878.35 ± 177.93 µg/40 mL) times more EVs proteins than bacteria cultured without LZD (379.55 ± 43.07 µg/40 mL) (Fig. 6a). DLS results showed that LZD did not affect the EVs particle size (Fig. 6b). Moreover, SDS-PAGE analysis showed similar protein profiles among EVs of Efm cultured without or with LZD respectively (Fig. 6c). These results suggested that 1/10 MIC of LZD stimulated EVs secretion without affecting their protein composition.

### Increased EVs secretion is beneficial for the reduction of gastric mucosal ulcer, histological lesions, and mucosal glycoprotein lessen

Then, we assessed whether the intragastric coadministration of Efm and LZD can improve the gastroprotective effect in rat. The schematic diagram of the experimental procedures was shown in Fig. 7a. SD rats were pretreated with LZD, Efm, or coadministration of Efm and LZD every other day for a total of three times before ethanol treatment. Macroscopic evaluation of gastric injury were shown in Fig. 7b. The results showed that coadministration of Efm and LZD improved the gastroprotective effect compared to the other two groups, there were no significant differences between the three groups ($P > 0.05$). Compared with the Efm group, UI in the Efm-LZD group was decreased (UI, 2.88 ± 2.34 vs. 3.88 ± 2.03, $P > 0.05$). Histopathological analysis of gastric mucosa stained with H&E were shown in Fig. 7c. LZD group micrograph indicated mild pathological changes including bleeding (✕) and epithelial damage (★). Efm group micrograph indicated moderate pathological changes. Efm-LZD group micrograph depicted normal gastric mucosal histology. Sections were also stained with PAS (Fig. 7d) to evaluate mucosal glycoprotein production. Efm-LZD group showing stronger PAS reaction among surface mucous and neck cells (▲) LZD and Efm group. Considered together, these results suggested that increased EVs secretion is beneficial for the gastroprotective effect of Efm.

### Increased EVs secretion reduced gastric pH, H+/K+-ATPase activity and improved antioxidant level

We further assessed gastric pH, H+/K+-ATPase activity and antioxidant level. As shown in Fig. 8a. Efm and LZD coadministration displayed lowest gastric pH, gastric tissues H+/K+-ATPase, MDA, MPO, and highest GSH-Px, GSH levels among three groups. Efm and LZD coadministration significantly reduced gastric pH level, and increased gastric tissues GSH-Px level compared to Efm group ($P < 0.05$). These results suggested that increased

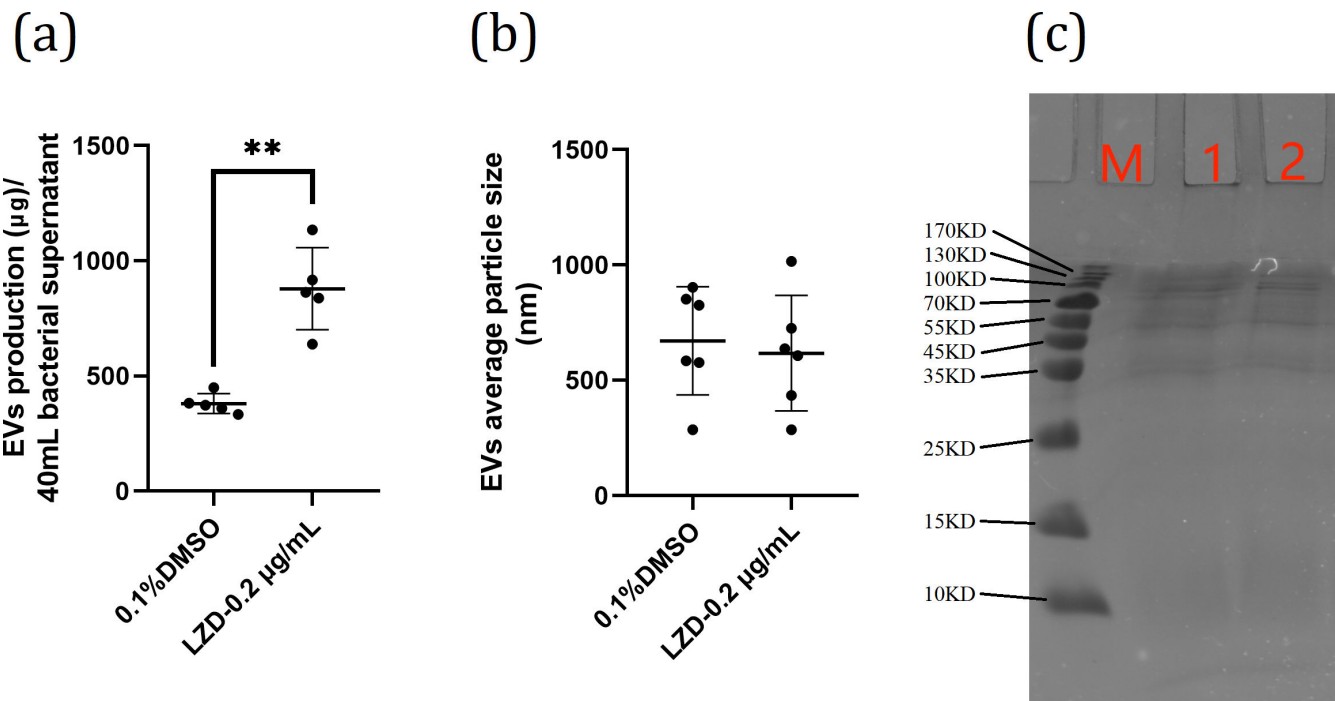

**FIG 6** Effects of LZD on Efm EVs secretion. (a) Production of EVs from Efm cultured with or without LZD. EVs were isolated from Efm cultured in 40 mL MRS medium with 0.2 µg/mL LZD or not. The protein concentration of EVs isolated from bacterial culture was measured using a modified BCA assay. Data are expressed as mean ± SD. ($n = 6$; **$P < 0.01$, Student's $T$ test.). (b) The average particle size of EfmEVs measured by DLS. Data are expressed as mean ± SD. ($n = 6$; **$P < 0.01$, Student's $T$ test.). (c) SDS-PAGE analysis of EVs proteins. Lane M, molecular weight marker; 1, EVs without LZD; 2, EVs with LZD.

EVs secretion ameliorated ethanol-induced gastric injury via the inhibition of gastric acid secretion and the enhancement of mucosal antioxidant level.

### Increased EVs secretion improved gastric mucin secretion proteins and inflammatory related markers mRNA expression

We next quantified the expression of vasoconstrictor (ET-1), mucin secretion proteins (Muc1, Muc6), and inflammatory related markers (NF-κb1, IL-1b, IL-6, and IL-10) in the gastric samples by RT-PCR. As shown in Fig. 8b, Efm and LZD coadministration displayed highest ET-1 and Muc6 and lowest IL-1b, IL-6, and IL-10 mRNA expression among three groups. Efm and LZD coadministration significantly increased gastric tissue ET-1 mRNA expression compared with the Efm group ($P < 0.05$). These results suggested that increased EV secretion ameliorated ethanol-induced gastric injury via the enhancement of mucosal barrier function and the inhibition of the proinflammatory response.

### Increased EV secretion improved the serum antioxidant level

We further assessed the serum antioxidant level and inflammatory factor characteristics. As shown in Fig. 8c, Efm and LZD coadministration displayed lowest MDA and ET-1 and highest GSH-Px, GSH, NO, MPO, and TNF-α contents in serum among three groups. Efm and LZD coadministration significantly increased serum GSH-Px and GSH contents compared with the Efm group ($P < 0.05$). Results suggested that increased EV secretion ameliorated ethanol-induced gastric injury via the enhancement of the serum antioxidant level.

## DISCUSSION

Gastric injury occurs due to an imbalance between the protective and aggressive factors in the gastric mucosa (34). The gastroprotective factors include antioxidants,

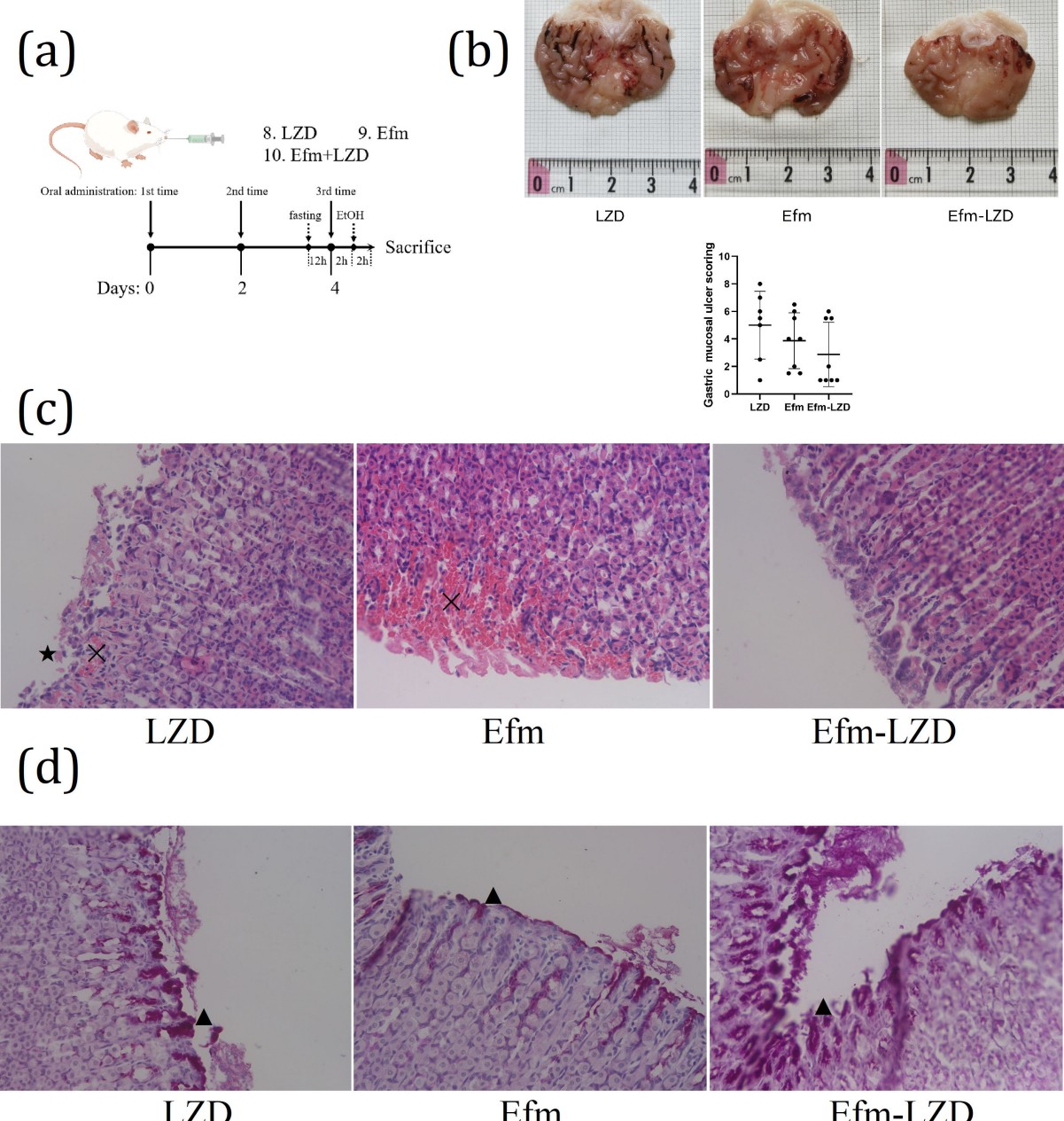

**FIG 7** Protective effect of Efm was improved by increasing EVs secretion in rat. (a) Experimental strategy used for the animal experiments. To investigate whether EVs secretion is beneficial for the gastroprotective effect, rats were divided into LZD, Efm and Efm-LZD groups ($n$ = 8/group). The rats in the LZD, Efm and Efm-LZD groups were, respectively, gavaged with 0.2 µg/mL of LZD (volume 5 mL/kg BW), $2 \times 10^9$ CFU/mL of Efm (volume 5 mL/kg BW), and Efm mixed with LZD ($2 \times 10^9$ CFU/mL of Efm, 0.2 µg/mL of LZD, volume 5 mL/kg BW) every other day for a total of three times. Before the final gavage, rats were fasted for 12 h with free access to water. Two hours after the final gavage, the rats were gavaged with a single dose of absolute ethanol (5 mL/kg BW) to induce acute gastric mucosal injury. (b) Macroscopic evaluation of gastric injury. The bar graphs show gastric mucosal ulcer index determined by morphological analysis. (c) Histopathological analysis of gastric mucosa (H&E stain, ×40). (d) Photomicrographs showing different reactions among mucosal layer in stomach sections in different groups (PAS stain ×400). Data are expressed as mean ± SD. Means with different letters were significantly different ($n$ = 8; One-Way ANOVA; Duncan; $P < 0.05$).

bicarbonate, mucin, NO, and prostaglandins in the gastric mucosa. The damaging factors include oxidative stress, inflammatory response, gastric acid, *Helicobacter pylori*, and NSAIDs (35). Ethanol is an important contributor to gastric mucosal lesions and is widely used to induce gastric mucosal injury in the animal models (4, 36). Ethanol intake directly damages the gastric mucosa, resulting in gastric mucosal bleeding, ulcers,

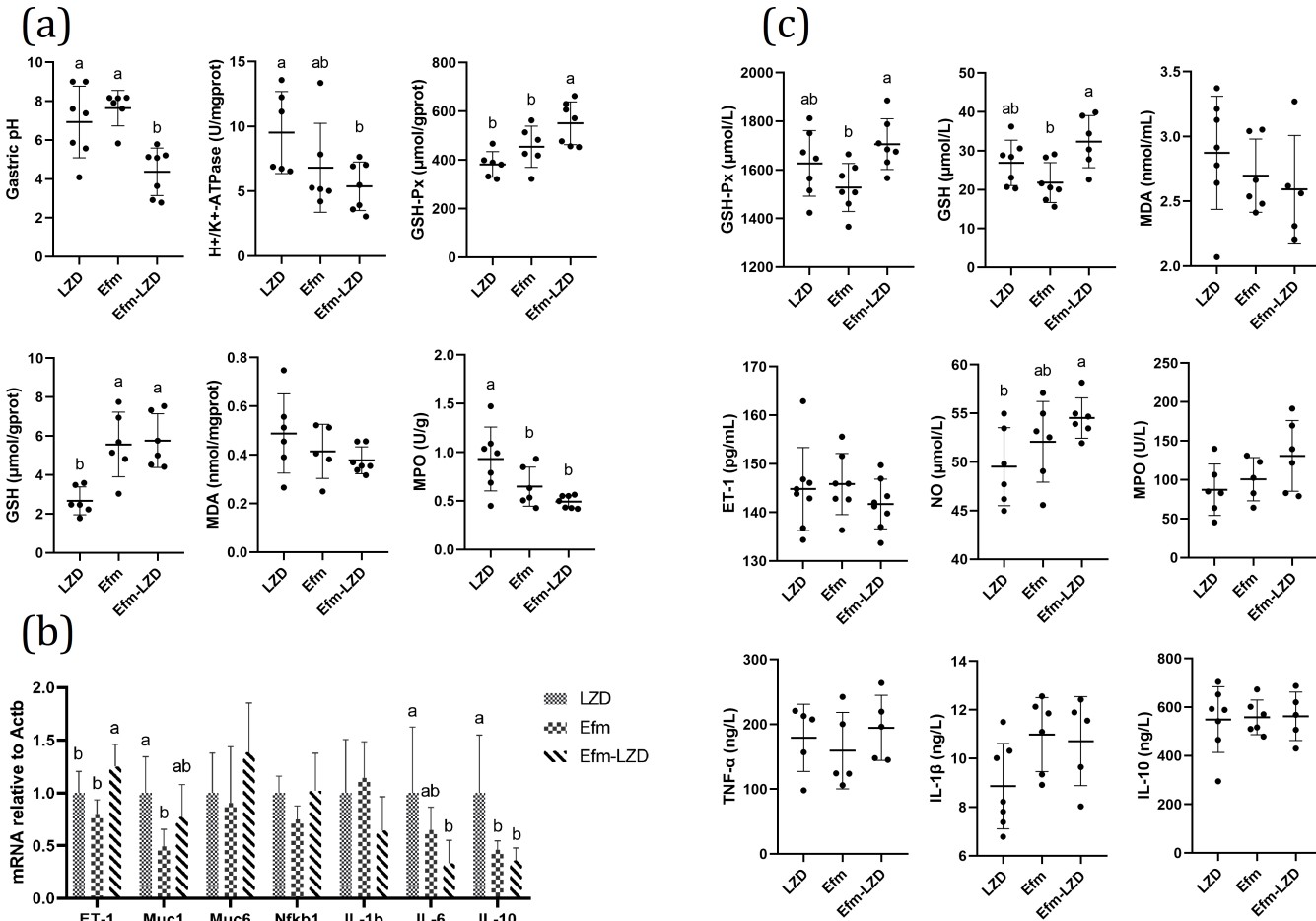

**FIG 8** Effects of increasing EVs secretion on gastric pH, biochemical indexes and mRNA expression in rat. (a) Gastric pH and the contents of H+/K+-ATPase, GSH-Px, GSH, MDA, MPO in gastric tissues. (b) Relative mRNA expression levels of *ET-1*, *Muc1*, *Muc6*, *Nfkb1*, *IL-1b*, *IL-6* and *IL-10* in the gastric tissues. (c) The contents of GSH-Px, GSH, MDA, ET-1, NO, MPO, TNF-α, IL-1β and IL-10 in serum. Data are expressed as mean ± SD. Means with different letters were significantly different ($n = 8$; One-Way ANOVA; Duncan; $P < 0.05$).

erosion, overproduction of gastric juice, oxidative stress, and inflammation (5). Probiotics are well known for many health effects on the host when consumed, especially on the gastrointestinal tract (6). Many studies have reported the gastroprotective effect of probiotic microorganisms (3–5). In this study, we determined the effect of Efm on ethanol-induced gastric injury in rats. Macroscopic evaluation and H&E staining results showed that ethanol (5 mL/kg BW) orally administered resulted in severe bleeding and ulcer, and Efm pretreatment markedly attenuated gastric injury.

Mucus is the first line of defense barrier of the gastric mucosa against injury and into which bicarbonate is secreted by the underlying epithelial cells. The "mucus-bicarbonate" barrier sustains a pH gradient between the lumen and cell surface such that epithelial cells are maintained at a suitable pH environment (37). H+/K+-ATPase is a key enzyme in gastric acid production and promotes the overproduction of gastric juice and resulting gastric injury directly (38). In this study, alcohol induced an increase in the activity of H+/K+-ATPase (Fig. 2a) and a reduce in *Muc1* and *Muc6* mRNA expression (Fig. 2b) and mucosal glycoprotein production (Fig. 1d), which was consistent with previous research reports (2, 38). Efm pretreatment can protect the stomach from ethanol damage by recovering H+/K+-ATPase activity and *Muc1* and *Muc6* mRNA expression and mucosal glycoprotein production.

Oxidative stress plays a significant role in the pathogenesis of ethanol-induced gastric injury, and GSH-Px and MDA were measured as biomarkers for the oxidative stress. In

this study, alcohol induced reduction in serum GSH-Px level (Fig. 2c) and an increase in gastric and serum MDA levels (Fig. 2a and c), which was consistent with previous research reports (5). Efm pretreatment recovered GSH-Px and MDA levels ideally. In addition, inflammation response is the most direct manifestation of gastric injury. Previous studies reported that the higher expressions of some proinflammatory genes were observed in gastric tissue after ethanol treatment (30, 39). TNF-α, IL-1β, and IL-6 are proinflammatory cytokines (40). Increased IL-1β could recruit TNF-α in the process of gastric injury (41). TNF-α activates neutrophil infiltration to cause the disturbance in gastric microcirculatory, thereby aggravating gastric injury (4). Furthermore, MPO is an indicator of neutrophil infiltration (42). IL-10 serves as an important regulator in preventing proinflammatory responses by regulating over-exuberant immune responses and autoimmune pathologies (43). In the present study, alcohol induced an increase in gastric *IL-1b*, *IL-6*, *IL-10* mRNA expression (Fig. 2b), serum TNF-α, IL-1β, MPO (Fig. 2c), and gastric MPO levels (Fig. 2a), which was consistent with previous research reports. Efm pretreatment recovered gastric *IL-1b*, *IL-6* mRNA expression, serum TNF-α, IL-1β, MPO , and gastric MPO levels ideally, indicating that inflammation response was alleviated.

Based on the above results, we determined the gastroprotective effect of Efm. In order to explore the gastroprotective effect substance basis of Efm, bacterial culture was roughly classified into three groups: bacterial supernatants (inactivated by heat, iaEfm), EVs isolated from the cultured supernatants (EVs), and EVFS. Gastroprotective effects were compared between the three components; it was found that EV pretreatment showed the smallest injury (Fig. 4b); mild pathological changes (Fig. 4c); more glycoprotein production (Fig. 4d); lowest gastric pH, gastric tissue H+/K+-ATPase, MDA, and MPO levels (Fig. 5a); highest *Muc1* and *Muc6* expression; lowest *IL-1b*, *IL-6*, and *IL-10* mRNA expression (Fig. 5B); highest NO; and lowest MDA, TNF-a, IL-1β, and IL-10 contents in serum. Furthermore, LZD-stimulated EV secretion is beneficial for the gastroprotective effect of Efm, with the specific manifestation of smallest injury (Fig. 7b), normal gastric mucosal histology (Fig. 7c), more glycoprotein production (Fig. 7d), reduced gastric pH, inflammatory response, and enhanced antioxidant ability (Fig. 8a through c). These results showed that EVs are an important functional active component of Efm gastroprotective effect.

Many studies have suggested the close relationship between biodistribution and function of exogenous EVs in animals. Ginger EVs mainly target the gut and were taken up by gut microbes, thereby altering gut microbial composition and host physiological function (44). *Lactobacillus paracasei*-derived EVs were mainly distributed in the gastrointestinal tract and attenuate LPS-induced inflammation in the intestine through ER stress activation (21). *Helicobacter pylori*-derived EVs specifically target gastric mucosa and induce the production of TNF-α, IL-6, and IL-1β by macrophages and IL-8 by gastric epithelial cells (45). In this study, EfmEVs mainly target the stomach and intestine of mice, which strongly implies an important role in the regulation of gastrointestinal function and was confirmed in our study.

Bacterial EVs have great prospects for application and development because they can be more easily customized and can be produced in large quantities by using bacterial fermentation vessels (46). The anti-inflammaging and antioxidant effects of EfmEVs suggest their potential as modulators of cell senescence. However, what are the functional molecules in EfmEVs? Whether functional molecules can be engineered to further enhance or specialize the function of EfmEVs is worthy of further study.

The present study has some potential limitation. Firstly, the average mucosal ulcer scoring (UI) showed some variance between the different batches of rats with the same treatment (UI for Efm in Fig. 1b and 7b). UI of rats with different treatments and batches showed similar results (UI for Efm and iaEfm in Fig. 1b and 4b). This could be due to random error and inherent variability in the assay between experiments. Secondly, the effect of sub-minimum inhibitory concentration LZD on Efm is not fully understood. In addition to increasing the amount of EV secretion, did LZD (sub-MICs) cause changes in the active components of EVs and the properties of Efm. Further prospective studies

should be designed to overcome this limitation. Thirdly, PKH or Dir dyes have been widely used for labeling extracellular vesicles. However, recent studies have shown that nanoparticles of dye can also be internalized independently. With the advancement of research methods, further research on this question is warranted.

## Conclusions

In conclusion, *Enterococcus faecium* significantly relieved ethanol-induced gastric injury in rats and EVs are an important functional active component. This beneficial effect may be through an anti-inflammatory and anti-oxidative mechanism by improving mucus secretion and the biosynthesis of GSH-Px and NO, reducing the levels of TNF-α, MPO, and MDA, downregulating the expressions of proinflammatory genes (*IL-1β* and *IL-6* mRNA).

## ACKNOWLEDGMENTS

This work was supported by the National Natural Science Foundation of China (Grant No. 31902228), Wenchang Chicken superiority characteristic industrial cluster project (Grant No. WCSCICP20211106), Discipline Construction Program of Foshan University (Grant No. CGZ0400162), Special Fund for Science and Technology Innovation Cultivation of Guangdong University Students (Grant No. pdjh2024b397), and Basic and Applied Basic Research Foundation of Guangdong Province (Grant Nos. 2019A1515110780 and 2023A1515010596).

Conceptualization was done by Q.X. and Y.Z.; data curation was done by J.S.; formal analysis was done by S.L.; funding acquisition was done by Q.Q., L.W., H.Z., and X.F.; investigation was done by M.L.; methodology was done by Q.Q.; project administration was done by Q.Q.; resources were acquired by L.W.; software was acquired by R.S.; supervision was done by H.Z.; validation was done by Q.Q.; visualization was done by T.C.; writing—original draft—was done by M.L.; writing—review and editing—was done by Q.Q. and X.F.; all authors have read and agreed to the published version of the manuscript.

## AUTHOR AFFILIATIONS

[1]School of Life Science and Engineering, Foshan University, Foshan, China

[2]Sanya Institute, Hainan Academy of Agricultural Sciences (Hainan Experimental Animal Research Center), Sanya, China

[3]Institute of Animal Science and Veterinary Medicine of Hainan Academy of Agricultural Sciences, Haikou, China

[4]Guangdong Provincial Key Laboratory of Animal Nutrition Control, College of Animal Science, South China Agricultural University, Guangzhou, China

## AUTHOR ORCIDs

Qien Qi http://orcid.org/0000-0003-0080-9355

## FUNDING

| Funder | Grant(s) | Author(s) |
| --- | --- | --- |
| MOST | National Natural Science Foundation of China (NSFC) | 31902228 | Qien Qi |
| GDSTC | Basic and Applied Basic Research Foundation of Guangdong Province (廣東省基礎與應用基礎研究專項資金) | 2019A1515110780 | Xin Feng |
| GDSTC | Basic and Applied Basic Research Foundation of Guangdong Province (廣東省基礎與應用基礎研究專項資金) | 2023A1515010596 | Huihua Zhang |
| Wenchang Chicken superiority characteristic industrial cluster project | WCSCICP20211106 | Limin Wei |

| Funder | Grant(s) | Author(s) |
|---|---|---|
| Discipline Construction Program of Foshan University | CGZ0400162 | Huihua Zhang |
| Special Fund for Science and Technology Innovation Cultivation of Guangdong University Students | pdjh2024b397 | Meiying Luo |

## ETHICS APPROVAL

Animal care and experimental procedures were approved by the Animal Care and Use Committee of Foshan University (FOSU2104-1), which meet the ethical standards in Laboratory Animal—Guideline for Ethical Review of Animal Welfare (The National Standard of the People's Republic of China GB/T 35892-2018).

## ADDITIONAL FILES

The following material is available online.

### Supplemental Material

**Supplemental Tables (Spectrum03894-23-s0001.docx).** Microbial species identification results and sequences of primers for real-time PCR.

### Open Peer Review

**PEER REVIEW HISTORY (review-history.pdf).** An accounting of the reviewer comments and feedback.

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
