## [Reviewer comments · Microbiology Spectrum]

Microbiology Spectrum

Protective effect of *Enterococcus faecium* against ethanol-induced gastric injury via Extracellular Vesicles

Meiying Luo, Junhang Sun, Suqian Li, Limin Wei, Ruiping Sun, Xin Feng, Huihua Zhang, Ting Chen, Qianyun Xi, Yong-Liang Zhang, and Qien Qi

Corresponding Author(s): Qien Qi, Foshan University

Review Timeline:

Submission Date:	November 8, 2023
Editorial Decision:	December 18, 2023
Revision Received:	January 9, 2024
Editorial Decision:	January 30, 2024
Revision Received:	February 5, 2024
Accepted:	March 4, 2024

Editor: Tao Deng

Reviewer(s): Disclosure of reviewer identity is with reference to reviewer comments included in decision letter(s). The following individuals involved in review of your submission have agreed to reveal their identity: Lysangela Ronalte Alves (Reviewer #2)

Transaction Report:

DOI: <https://doi.org/10.1128/spectrum.03894-23>

Re: Spectrum03894-23 (Protective effect of *Enterococcus faecium* against ethanol-induced gastric injury via Extracellular Vesicles)

Dear Dr. Qien Qi:

Thank you for the privilege of reviewing your work. I have received comments from two reviewers. They all think your manuscript need extensive improving from various aspects. I would reconisder your muanscript if you could revise the manuscript extensively as suggested by the two reviewers. Below you will find the comments and instructions from the Spectrum editorial office.

Revision Guidelines

Sincerely,
Tao Deng
Editor
Microbiology Spectrum

Reviewer #1 (Comments for the Author):

In this manuscript, the authors present data suggesting that extracellular vesicles derived from the gram-positive bacteria *E. faecium* provide some protection against ethanol-induced gastric ulcer in rats. The authors measure multiple parameters, as well as gene expression of some mRNAs relevant to gastric ulcer. They then go on to show that increasing the number of

extracellular vesicles artificially using LZD also increase this protective effect.

Generally, the manuscript provides data to support the main conclusions, but little is done to put these results any context. The results section reads more like a figure legend and is challenging to assess due to limited information. The authors use the discussion section to give the results some context, but they do not present any potential limitations of the study. A few ideas on additional challenges that could be discussed are suggested below:

Major Comments

1. The text of the results is very fragmented. I appreciate the direct approach to relaying only the findings, but it is too simplistic. It makes understanding this section very difficult for the reader. Additional effort to explain the rationale and setup for each experiment would clarify the authors' motivations.
2. Line 112-114, it is unclear how the mass of EfmEVs delivered compares to the mass of EVFS or iaEfm used. Is it possible that the effects observed are dependent on the total material delivered? Can the authors comment on whether these amounts are physiologically relevant to the EVs naturally released by the bacteria or would this only be relevant in the case of artificial delivery. Relatedly, the scoring for Efm and iaEfm (Figure 1B and Figure 4B), appear to be similar. Does this argue against EV release as an active process? Similarly, when comparing Figure 1B vs 7B, the Efm scoring is quite different. I understand that there is some inherent variability in the assay between experiments, but this should be brought up in the discussion section as a potential limitation of the study.
3. Figure 1A (and 4A, 7A), I find this schematic a little confusing. What are the numbered items? Why is PBS listed twice? A few extra words in the legend might help.
4. Line 187-195, why were these genes chosen for measurement of mRNA expression? This information is at least partially included in the discussion, but it would help to have some rationale for the experimental setups with the results.
5. Line 258-269, it is difficult to assess what is happening in these experiments from these results descriptions. See comment 1 above.
6. Figure 5B, the error bars appear to be cut off by the y-axis break.
7. Can the authors rule out that they are not just delivering protein aggregates with their EV isolation method? Would they see the same effect if another EV isolation method like size-exclusion chromatography was used to isolate EVs? In this regard, Figure 6A would be strengthened by an alternative measure of EVs, such as NTA or TRPS that directly measures particles.
8. A related potential experiment would be to first treat the EVs with a low amount of detergent to see if breaking the EVs prevents the effect on gastric ulcer. A proper control of a small amount of detergent alone would also be required to make this meaningful. Have the authors considered such an experiment? Maybe it is not possible with rats.
9. What is the mechanism of action of LZD? Are there other effects on the bacteria that could be contributing to protection? Again, this could be discussed in the potential limitations of this study.
10. Are all EVs resistant to low pH or is this specific to gram positive bacterial EVs? Relatedly, if you provided EVs in high numbers from other gram-positive bacteria, would you also expect protection, or is this effect unique to *E. faecium*?

Minor Comments

1. The manuscript has a lot of acronyms, and many are first defined in the methods, which makes understanding the results section quite challenging at times. It would be helpful to reduce the total number of acronyms, or at least strategically redefine some in the results section. Some are only defined in the list of acronyms at the end (NIR, UI, etc).
2. Line 48-50, it could be beneficial to distinguish between bacterial-derived EVs and host derived EVs (exosomes, microvesicles).
3. Line 87, Linezolid mechanism of action should be defined at first description.
4. Line 170, The positive control group is denoted OME in the methods. Maybe it would be helpful to the reader to define OME alongside omeprazole? Some sort of description of omeprazole should also be included.
5. Figure 2B, the authors should consider switching to using colors that are friendly to color-blind readers.

Reviewer #2 (Comments for the Author):

In the manuscript entitled "Protective effect of *Enterococcus faecium* against ethanol-induced gastric injury via Extracellular Vesicles" Luo and co-authors studied the protective effect of the Extracellular vesicles produced by *E. faecium* compared the organism itself, and omeprazole, in the gastric injury model caused by ethanol. The observations made in the study are interesting, as this organism is part of our microbiota and could play an important role in preventing damage to the gastrointestinal tissue.

The authors used histological data and transcript expression to determine the level of injury observed in the stomach after different treatments and controls were used. The study draws attention to the role of EVs in this scenario. The authors used PBS and Omeprazole as controls, as well as the use of the supernatant after EV isolation. The EVs characterization also showed the MET and LDS for measurement and phenotype analysis.

Major comments:

The English needs extensive revision.

Lines 30 and 31 - "Gastric ulcer is the most common gastrointestinal disorder affects 10% of the world population with different

etiologies[1]. It is mucosal erosions caused by many factors such as..."
Disorder THAT affects.

"It is mucosal erosions" - lacks a connector.

Line 36 - "with an important functions", remove the AN

Line 48 - "Extracellular vesicles (EVs) are lipid-bilayers produced by all domains of life" - after bilayers it lacks a noun.

Line 53 - "However role of EV"

These are some examples found only on page one. So, the authors need to review the language throughout the entire text.

Line 146 - How many technical and biological replicates were performed?

Results subsections - The results section titles are presented as a methodology. The authors should call attention to the main result presented here.

Line 207 - How many EV isolations did the authors prepare? Was this reproducible?

Line 343 - What does "reasonable request" mean? Open science currently does not support and even discourages this type of statement. It lacks transparency and reproducibility.

Reviewer #1 (Comments for the Author):

In this manuscript, the authors present data suggesting that extracellular vesicles derived from the gram-positive bacteria *E. faecium* provide some protection against ethanol-induced gastric ulcer in rats. The authors measure multiple parameters, as gene expression of some mRNAs relevant to gastric ulcer. They then go on to show that increasing the number of extracellular vesicles artificially using LZD also this protective effect.

Generally, the manuscript provides data to support the main conclusions, but little is done to put these results any context. The results section reads more like a figure and is challenging to assess due to limited information. The authors use the discussion section to give the results some context, but they do not present any potential limitations of the study. A few ideas on additional challenges that could be discussed are suggested below:

Reply: We thank the reviewer for recognizing our work.

Based on the comments of the reviewer, We have made a comprehensive logical arrangement and series of the results section and presented some potential limitation of the study in the discussion section.

Major Comments

1. The text of the results is very fragmented. I appreciate the direct approach to relaying only the findings, but it is too simplistic. It makes understanding this section very difficult for the reader. Additional effort to explain the rationale and setup for each experiment would clarify the authors' motivations.

Reply: We greatly appreciate this reviewer's above comments. The results section are revised according to the comment that make our manuscript much more easy-to-read.

2. Line 112-114, it is unclear how the mass of EfmEVs delivered compares to the mass of EVFS or iaEfm used. Is it possible that the effects observed are dependent on

the total material delivered? Can the authors comment on whether these amounts are physiologically relevant to the EVs naturally released by the bacteria or would this only be relevant in the case of artificial delivery. Relatedly, the scoring for Efm and iaEfm (Figure 1B and Figure 4B), appear to be similar. Does this argue against EV release as an active process? Similarly, when comparing Figure 1B vs 7B, the Efm scoring is quite different. I understand that there is some inherent variability in the assay between experiments, but this should be brought up in the discussion section as a potential limitation of the study.

Reply: We greatly appreciate this reviewer's above comments. Due to our carelessness, we made mistakes in the calculation and description of the mass of EfmEVs delivered compares to the mass of EVFS or iaEfm. After the calculation confirmed, the original manuscript is revised to "In our experiment, each 1 mL overnight cultures contained about 2×10^9 CFU Efm, and 20 μ g EfmEVs. Therefore, 1 mL EVFS was comparable to 2×10^9 CFU inactivated Efm and 20 μ g EfmEVs in the subsequent tests." , Line 115-116.

These amounts of Efm and EfmEVs in a given volume was the experimental results of the author in vitro culture. In the natural environment, due to differences in nutritional conditions, bacterial density, and other environmental conditions, this may have an impact.

We strongly agree with the reviewers' understanding of the differences of scoring in different experiments. Following the reviewer's suggestion, the author brought up this limitation in the discussion section, Line 365-368.

3. Figure 1A (and 4A, 7A), I find this schematic a little confusing. What are the numbered items? Why is PBS listed twice? A few extra words in the legend might help.

Reply: Three animal tests were conducted in this study, with a total of 11 treatment groups. The item number represents the treatment group number in the full text.

PBS listed twice in Figure 1a. In the first animal test, rats in the normal control group were orally administered with 5 mL/kg body weight (BW) of PBS every other day for

a total of three times, and received 5 mL/kg BW of PBS to induce gastric mucosal injury at 2 h after the last treatment.

The author has improved the description of the Figure legends.

4. Line 187-195, why were these genes chosen for measurement of mRNA expression? This information is at least partially included in the discussion, but it would help to have some rationale for the experimental setups with the results.

Reply: Endothelin-1 (ET-1) is a potent vasoconstrictor,

The gastric mucin can protect the gastric epithelium from diverse chemical or physical stimuli. Twelve genes encoding mucins have been reported. MUC1, MUC5AC, and MUC6 are included.

The nuclear transcription factor Nfkb1 plays a vital role in regulating the immune responses. IL-1b, IL-6, IL-10 are important inflammation - related cytokine in the body, mainly regulate the cell function and involved in the body immunity.

These genes are known to be activated in EtOH-induced gastric ulcer in mice.

The results section are revised according to the comment, Line 201-202.

5. Line 258-269, it is difficult to assess what is happening in these experiments from these results descriptions. See comment 1 above.

Reply: The results section are revised according to the comment 1 that make our manuscript much more easy-to-read.

6. Figure 5B, the error bars appear to be cut off by the y-axis break.

Reply: Revised.

7. Can the authors rule out that they are not just delivering protein aggregates with their EV isolation method? Would they see the same effect if another EV isolation method like size-exclusion chromatography was used to isolate EVs? In this regard, Figure 6A would be strengthened by an alternative measure of EVs, such as NTA or TRPS that directly measures particles.

Reply: The chemical components of Efm-EVs were analyzed by multi-omics method. The authors found that in addition to proteins, Efm-EVs also contained RNA, metabolites and other components. The BCA for total proteins yield was most used to reported EVs quantification.

The authors conducted EVs isolation by using differential ultracentrifugation (dUC) consistently. Therefore, the authors are uncertain about the effectiveness of EV isolation method.

EVs average particle size measurement result are added to Figure 6B. The results showed no difference in EVs size.

8. A related potential experiment would be to first treat the EVs with a low amount of detergent to see if breaking the EVs prevents the effect on gastric ulcer. A proper control of a small amount of detergent alone would also be required to make this meaningful.

Have the authors considered such an experiment? Maybe it is not possible with rats.

Reply: Thank you for the constructive comment. We will consider designing this experiment in future studies.

9. What is the mechanism of action of LZD? Are there other effects on the bacteria that could be contributing to protection? Again, this could be discussed in the potential limitations of this study.

Reply: Linezolid is a synthetic antibiotic which prevents the synthesis of bacterial protein via binding to rRNA on both the 30S and 50S ribosomal subunits. Linezolid is a synthetic antibiotic which prevents the synthesis of bacterial protein via binding to rRNA on both the 30S and 50S ribosomal subunits. Studies have reported that the MIC of LZD against Efm is 2 µg/mL (DOI: 10.3389/fcimb.2019.00295). In this paper, 0.2 µg /mL LZD was used to induce EVs secretion according to previous reports. The mechanism of sub-minimum inhibitory concentrations (MICs) LZD-induced EVs secretion is not fully understood. Following the reviewer's suggestion, the author brought up this limitation in the discussion section (Line 365-371).

10. Are all EVs resistant to low pH or is this specific to gram positive bacterial EVs? Relatedly, if you provided EVs in high numbers from other gram-positive bacteria, would you also expect protection, or is this effect unique to *E. faecium*?

Reply: We don't really know, and more research is needed to shed light on this issue. Our study only describes the protective effect of EfmEVs against gastric injury, and does not involve other gram positive bacterial EVs.

Minor Comments

1. The manuscript has a lot of acronyms, and many are first defined in the methods, which makes understanding the results section quite challenging at times. It would be helpful to reduce the total number of acronyms, or at least strategically redefine some in the results section. Some are only defined in the list of acronyms at the end (NIR, UI, etc).

Reply: Thank you for your comment. We redefined acronyms in the results section accordingly.

2. Line 48-50, it could be beneficial to distinguish between bacterial-derived EVs and host derived EVs (exosomes, microvesicles).

Reply: Revised accordingly (Line 51-55).

3. Line 87, Linezolid mechanism of action should be defined at first description.

Reply: Revised accordingly (Line 92-93).

4. Line 170, The positive control group is denoted OME in the methods. Maybe it would be helpful to the reader to define OME alongside omeprazole? Some sort of description of omeprazole should also be included.

Reply: Revised accordingly (Line 179 and Figure 1 legend).

5. Figure 2B, the authors should consider switching to using colors that are friendly to

color-blind readers.

Reply: Thank you for your comment. I'm sorry we didn't take that into account at first. Figure 2B, Figure 5B, and Figure 8B have been revised accordingly.

Reviewer #2 (Comments for the Author):

In the manuscript entitled "Protective effect of *Enterococcus faecium* against ethanol-induced gastric injury via Extracellular Vesicles" Luo and co-authors studied the protective effect of the Extracellular vesicles produced by *E. faecium* compared the organism itself, and omeprazole, in the gastric injury model caused by ethanol. The observations made in the study are interesting, as this organism is part of our microbiota and could play an important role in preventing damage to the gastrointestinal tissue.

The authors used histological data and transcript expression to determine the level of injury observed in the stomach after different treatments and controls were used. The study draws attention to the role of EVs in this scenario. The authors used PBS and Omeprazole as controls, as well as the use of the supernatant after EV isolation. The EVs characterization also showed the MET and LDS for measurement and phenotype analysis.

Reply: Thank you for your recognition of our work.

Major comments:

The English needs extensive revision.

Reply: The entire text has been reviewed accordingly.

Lines 30 and 31 - "Gastric ulcer is the most common gastrointestinal disorder affects 10% of the world population with different etiologies[1]. It is mucosal erosions caused by many factors such as..."

Disorder THAT affects.

"It is mucosal erosions" - lacks a connector.

Reply: Revised (Line 30-31).

Line 36 - "with an important functions", remove the AN

Reply: Revised (Line 36).

Line 48 - "Extracellular vesicles (EVs) are lipid-bilayers produced by all domains of life" - after bilayers it lacks a noun.

Reply: The original manuscript is revised to "lipid-bilayer vesicles" (Line 49).

Line 53 - "However role of EV"

Reply: The original manuscript is revised to "However, the role of EVs" (Line 58).

These are some examples found only on page one. So, the authors need to review the language throughout the entire text.

Reply: The entire text has been reviewed.

Line 146 - How many technical and biological replicates were performed?

Reply: More than twice technical replicates have been done for the main results of this study, the results of the preliminary tests on mice were not shown. Eight biological replications were used in the rat experiments, six biological replications were used in the stimulation of EVs secretion experiments, and it is explained in detail in the "experimental design" section (Line 107-126) and "Figure legends" section (Line 504-553).

Results subsections - The results section titles are presented as a methodology. The authors should call attention to the main result presented here.

Reply: The results section titles have been revised according to the comment.

Line 207 - How many EV isolations did the authors prepare? Was this reproducible?

Reply: We used 1 L overnight cultures for EVs isolation generally. About 20 mg EVs can be isolated from 1 L of cultures, which was described in detail in Line 115 of the manuscript. We strictly perform every step of the EVs isolation process to ensure repeatability of the test.

Line 343 - What does "reasonable request" mean? Open science currently does not support and even discourages this type of statement. It lacks transparency and reproducibility.

Reply: Thank you for your advice. The original manuscript is revised to “The datasets used and analysed during the current study are available from the corresponding author” (Line 393).

Re: Spectrum03894-23R1 (Protective effect of *Enterococcus faecium* against ethanol-induced gastric injury via Extracellular Vesicles)

Dear Dr. Qien Qi:

Thank you for the privilege of reviewing your work. Below you will find my comments, instructions from the Spectrum editorial office, and the reviewer comments. Your manuscript still need further editing and improving.

Revision Guidelines

Sincerely,
Tao Deng
Editor
Microbiology Spectrum

Reviewer #1 (Comments for the Author):

The manuscript is greatly improved over the first version, but additional editing would be beneficial. A few minor points are listed below.

1) I still do not understand the numbers listing the samples in Panel 1A (and subsequent panels in other figures). In the rebuttal, the authors directed us to the text, but I can find no mention of these values. I suspect that the PBS treated samples are with

and without ethanol? Maybe a little more editing of the legend would be helpful.

2) In Fig 3C, the authors show tracking of EV populations and state in the results that the EVs accumulate in the stomach. The dye also accumulates in the stomach, which should be mentioned. Many studies have shown that dyes can separate and traffic independently of EVs in vivo. This should potentially be mentioned as a potential caveat of this experiment, unless the authors have additional data to support their conclusion.

3) I agree with reviewer two that the datasets should be made publicly available, as stated by the journal.

4) Line 174, "lessen" should be "lesion" possibly?

Reviewer #2 (Comments for the Author):

The authors addressed the comments and questions accordingly. Figure 6 didn't include the ladder sizes in the protein gel.

Reviewer #1 (Comments for the Author):

The manuscript is greatly improved over the first version, but additional editing be beneficial. A few minor points are listed below.

1) I still do not understand the numbers listing the samples in Panel 1A (and panels in other figures). In the rebuttal, the authors directed us to the text, but I can no mention of these values. I suspect that the PBS treated samples are with and ethanol? Maybe a little more editing of the legend would be helpful.

Reply: Thank you for your further advice. Legends in Figures 1(a), 4(a) and 7(a) have been modified accordingly.

2) In Fig 3C, the authors show tracking of EV populations and state in the results that the EVs accumulate in the stomach. The dye also accumulates in the stomach, which should be mentioned. Many studies have shown that dyes can separate and traffic independently of EVs in vivo. This should potentially be mentioned as a potential caveat of this experiment, unless the authors have additional data to support their conclusion.

Reply: Thank you for your advice. This question is mentioned as a potential limitation in the discussion section.

3) I agree with reviewer two that the datasets should be made publicly available, as stated by the journal.

Reply: Thank you for your advice. This study did not involve sequencing data, but mainly consisted of original pictures and data records during the experiment. The original manuscript is revised to “The datasets used and analysed during the current study are available from the corresponding author” in R1 manuscript (Line 393).

4) Line 174, "lessen" should be "lesion" possibly?

Reply: Revised.

Reviewer #2 (Comments for the Author):

The authors addressed the comments and questions accordingly. Figure 6 didn't include the ladder sizes in the protein gel.

Reply: Revised accordingly (Figure 6c).

Re: Spectrum03894-23R2 (Protective effect of *Enterococcus faecium* against ethanol-induced gastric injury via Extracellular Vesicles)

Dear Dr. Qien Qi:

Your manuscript has been accepted, and I am forwarding it to the ASM production staff for publication. Your paper will first be checked to make sure all elements meet the technical requirements. ASM staff will contact you if anything needs to be revised before copyediting and production can begin. Otherwise, you will be notified when your proofs are ready to be viewed.

Sincerely,
Tao Deng
Editor
Microbiology Spectrum